# Rats Selected for Different Nervous Excitability: Long-Term Emotional–Painful Stress Affects the Dynamics of DNA Damage in Cells of Several Brain Areas

**DOI:** 10.3390/ijms25020994

**Published:** 2024-01-13

**Authors:** Veronika Shcherbinina, Marina Pavlova, Eugene Daev, Natalia Dyuzhikova

**Affiliations:** 1Laboratory of Higher Nervous Activity Genetics, Pavlov Institute of Physiology, Russian Academy of Sciences, nab. Makarova, 6, 199034 Saint Petersburg, Russia; sherbinina.veronika2014@yandex.ru (V.S.); pavlova.mb@gmail.com (M.P.); or e.daev@spbu.ru (E.D.); 2Department of Genetics and Biotechnology, Faculty of Biology, Saint Petersburg State University, Universitetskaya nab., 7–9, 199034 Saint Petersburg, Russia

**Keywords:** rats, stress, nervous system excitability, PTSD, compulsive disorder, DNA damage, comet assay, prefrontal cortex, hippocampus, amygdala

## Abstract

The maintenance of genome stability is critical for health, but during individual ontogenesis, different stressors affect DNA integrity, which can lead to functional and/or structural changes in the cells of target organs. In the nervous system, cell genome destabilization is associated with different neurological and psychiatric diseases, but experiments in vivo, where a link between stress and DNA instability has been demonstrated, are relatively rare. Here, we use rat strains selected for the contrast excitability of the tibialis nerve (*n. tibialis*) and nonselected Wistar rats to investigate the reasons for individual differences in developing post-stress pathologies. Previous research on the behavioral response of these strains to prolonged emotional–painful stress (PEPS) allows us to consider one strain as a model of post-traumatic stress disorder (PTSD) and another strain as a model of compulsive disorder (CD). We study DNA damage in the cells of the prefrontal cortex (PFC), hippocampus, and amygdala, regions involved in stress responses and the formation of post-stress dysfunctions. The evaluation of cell genome integrity via the comet assay shows different responses to PEPS in each brain area analyzed and for all strains used. This could help us to understand the reasons for individual differences in the consequences of stress and the pathophysiology of post-stress disease formation.

## 1. Introduction

The flexibility of a genome is an essential property that accompanies all ontogenesis. It is important for the adaptability and evolution of organisms. In somatic mosaicism, resulting from DNA alterations in neuronal progenitors, it increases the diversity of neurons and plays a role in normal brain physiology [1]. But, the increase in DNA damage level can lead to the dysfunction of cells and their death, which is one of the reasons for aging, cancer, neurological diseases, and neurodegeneration. The role of genome instability has been shown for a wide spectrum of disorders, including Parkinson’s and Alzheimer’s disease, Lewy body dementia, and amyotrophic lateral sclerosis [2].

There are many endogenous and exogenous inductors of DNA damage in central nervous system (CNS) cells: replication errors, spontaneous base deamination, abasic sites, oxidative damage, ionizing radiation, chemical mutagens, and other types of environmental stressors [3]. The stress reaction (the response to stressors of different natures, including psychoemotional) can be a time-consuming process that can lead to genome instability in the cells of target organs in definite stressor-sensitive stages of organism ontogenesis [4]. As a nonspecific process, stress responses activate the hypothalamus–pituitary–adrenal axis and increase glucocorticoid and catecholamine secretion, which are both associated with genome instability through changes in DNA repair systems [5,6], oxidative stress induction [7], and increases in the activity of transposons [8].

The individual differences in stress tolerance are dependent on the genotype, developmental specificity, or previous experience. One of the hereditary characteristics, which can be associated with different characteristics of stress response, is the level of excitability of the nervous system. To study the role of this characteristic in animal physiology and behavior in I. P. Pavlov Institute of Physiology of RAS, two rats’ strains with contrast excitability of *n. tibialis* were selected. These strains were comprehensively observed in different tests, both in physiological and post-stress conditions. The results showed strain-specific responses to PEPS and long-lasting effects (behavioral, morphological, genetic, and epigenetic), which can be found up to 6 months after the end of treatment. The strain with a high excitability threshold (HT, low excitable strain) after PEPS demonstrates depressive-like behavior; the strain with a low threshold of excitability (LT, high excitable strain) increases compulsive behavior. These changes in behavior and other characteristics allowed us to consider these strains as models of PTSD (HT strain) and CD (LT strain) in humans [9].

Different regions and structures are involved in stress responses and the formation of post-stress pathologies, which were comprehensively studied in rodent models [10,11]. The circuitry responsible for the adaptation to stressors includes the hippocampus, amygdala, and higher cortical areas (particularly the PFC). The disturbance in the functioning of this circuitry, induced by repeated stress, leads to cognitive violations and psychiatric disorders such as depression or PTSD [12]. The reaction of brain cells to each type of stress treatment is not the same in different brain structures (and in various regions of one structure) and depends on the cell types. It complicates the research of the genome response to stress in CNS cells and highlights the importance of single-cell methods to understand the mechanisms of disease occurrence and individual differences in the consequences of exposure to stressors.

By suggesting that altered excitability could influence different stress-involved brain structures dissimilarly, as well as cellular stress sensitivity, we tried to study immediate and delayed genome responses to PEPS in some brain regions. We used a comet assay (alkaline and neutral) to investigate the effect of PEPS on the DNA integrity of the PFC, hippocampus, and amygdala cells in rats with the contrast excitability of *n. tibiallis* (HT and LT strains), as well as in nonselected rats (Wistar strain). The results can reveal the role of brain cell genome instability in the formation of post-stress pathologies (PTSD and CD) in individuals with a hereditary difference in the excitability of the nervous system and the timing specificity of stress responses in different target brain regions.

## 2. Results

We studied both the immediate (1 or 2 h after the end stress procedure) and delayed (2 weeks and 2 months after stress) effects of PEPS on the cell genome integrity of three brain regions of each rat strain. The DNA content in comet tails (% of tDNA) was considered a DNA damage indicator for our study. Overall, 47,999 PFC cells, 34,163 hippocampal cells, and 42,332 amygdala cells were analyzed.

### 2.1. DNA Instability Dynamics in PFC Cells

Two hours after the last stress exposure, a change in the PFC stability of the cell genome of the PFC of HT and LT rats was shown. The changes found were opposite for LT and HT strains. In the case of the Wistar strain, no significant changes were found (Figure 1, Appendix A).

A delayed consequence of PEPS on the level of DNA damage in brain cells was studied in two post-stress periods: 2 weeks and 2 months. A significant elevation in tDNA content was shown only for the LT strain two weeks after the PEPS exposure (Figure 1, Appendix A). Two months after PEPS in PFC, no differences in the level of genome instability were demonstrated for the three strains used in the investigation (Figure 1, Appendix A).

### 2.2. DNA Instability Dynamics in Hippocampal Cells

In the hippocampus, only delayed consequences of PEPS on the level of DNA damage were studied. Our analysis of genome instability two weeks after stress did not reveal any differences in tDNA content for the three strains of rats (Figure 2, Appendix A). However, two months after PEPS, an increase in the amount of DNA damage was shown in the hippocampal cells of two strains, HT and Wistar (Figure 2, Appendix A), but the fold change in the median tDNA from the corresponding control was greater in the HT strain than in Wistar rats (1,88 and 1,53, correspondingly). For the LT strain, no difference in the level of genome instability in hippocampal cells two months after PEPS was found (Figure 2, Appendix A).

### 2.3. DNA Instability Dynamics in Amygdala Cells

For the HT and LT strains, an increased level of DNA damage was observed one hour after the last stress exposure (Figure 3, Appendix A), i.e., a unidirectional change was demonstrated (also close in terms of the change ratio relative to the corresponding controls: 1.54 times for the HT strain and 1.62 times for the LT strain). For the Wistar strain, the level of DNA damage in this post-stress period was not studied.

Research of the delayed consequences of PEPS in the amygdala also revealed some changes in the level of DNA damage in its cells. Delayed consequences of PEPS in amygdala cells appear as opposite interstrain changes (HT vs. Wistar) in DNA damage two weeks after PEPS (correspondingly decrease and increase) and absent for the LT strain (Figure 3, Appendix A). Two months after the end of PEPS in the amygdala, the tDNA content increased only for the HT strain, but for the LT and Wistar strains, no significant differences were shown (Figure 3, Appendix A).

### 2.4. DNA Dynamics Summary

Taken together, the results demonstrate the complex dynamics of DNA instability in brain cells after PEPS, which differ for all three studied strains (Figure 4). It is interesting that the changes in the level of DNA damage showed no correlation between two delayed post-stress periods—two weeks and two months after stress. In contrast, in all times when we found the increase in the level of DNA instability two months after PEPS, they were not preceded by an increased content of tDNA observed 1,5 months earlier (Figure 4B,C,H); if we found an increased level of DNA instability 2 weeks after stress, it was not followed by the same change in tDNA content 1,5 months later (Figure 4D,I). Inter-strain differences in the dynamics of post-stress genome instability were shown in all three examined brain structures: amygdala, PFC, and hippocampus (Figure 4). The strains used in this research had not only revealed a different amplitude of DNA damage but also different reparation abilities and, in some cases, even different directions of changes in genome response. In general, the amygdala and hippocampus of HT animals seem slower reacting toward PEPS because genome instability is revealed in two months. Researchers need to explore whether it could be from less effective cell genome reparation or a more severe selection of damaged neuronal cells in the LT strain. These results demonstrate the role of excitability—a hereditary-determined characteristic of the nervous system—in the response to stressful events and their immediate, long-lasting, or delayed consequences. Further studies are required to understand how the lack of effects in the LT strain at the genomic level of hippocampal and amygdala cells 2 months after the PEPS shown here correlates (positively or negatively) with the adaptation of the animal body.

## 3. Discussion

The ability to react quickly to environmental stressors and respond to them adequately is important to adapt to changing conditions and survive. But, the durations of stress reactions have not been long-lasting (or too strong) after the end of stressor actions because the overload of stress-targeted organs leads to their dysfunction and long-term post-stress pathologies. For example, even chronic mild stressors (CMSs) induce depressive-like behavior and pronounced structural and functional changes affecting the integrity of prefrontal GABAergic networks in Wistar rats [13]. The increase in DNA damage level is one of the signs that the organisms are failing to adapt to stressors using the regulatory mechanisms, i.e., in the reaction norm of its genotype, and it is also a risk factor for disease formation, as DNA repair is not always totally correct and effective.

Our results showed that, in the amygdala (Figure 4C,F) and hippocampus [13], right after (1 or 2 h) the PEPS ended, the level of genome damage increased for both the LT and HT strains. But, the long-term dynamics (2 weeks and 2 months after PEPS) of the DNA damage level are strain-specific. The low-excitable HT strain demonstrates a decrease in genome instability 2 weeks after stress to control or even lower levels both in the amygdala and hippocampus (Figure 4B,C). However, we found that in the amygdala and hippocampus of HT strains, 2 months after stress, the level of DNA damage increased again (Figure 4B,C). In contrast, in the high-excitable LT strain, no long-term effects of PEPS on genome instability were shown in the same brain structures (Figure 4E,F). Therefore, we demonstrate that the strain with a low threshold of nervous system excitability responds to PEPS with a greater increase in the DNA damage level in both the amygdala and the hippocampus [14] immediately (2 h) after the stress procedure ends but repairs these damages and does not induce them again after 2 weeks or 2 months. Although low-excitable strains (with a high threshold of excitability) react to stress with lower amplitudes of genome instability in the amygdala and hippocampus immediately after stress and repair the damages within 2 weeks, it also induces DNA instability de novo 2 months post-PEPS.

The distant damage of the hippocampus is considered an important element in the pathogenesis of cognitive and psychiatric disorders induced by focal brain injuries such as stroke or traumatic brain injury. The complication of these traumatic events is dependent not on the level of primary damage but on the level of biochemical response to them. The secretion of glucocorticoids after injury or exposure to stressors leads to neuroinflammation, neurodegeneration, and the disturbance of neurogenesis in the hippocampus, which are the reasons for the formation of secondary damages and pathologies [15]. Our results reveal that even emotional–painful stress can cause distant damage to the hippocampus and amygdala, a penchant for which is not correlated with the level of primary damage. The HT strain, where distant brain damage was shown, had a lower level of DNA instability immediately after stress than the LT strain, where distant DNA damage to the hippocampus and amygdala was not found. However, previous research on the distant effects of PEPS showed some delayed changes in the brain of LT strains: neuronal density in the CA3 region of the hippocampus decreased in two months after stress but not in 24 h or 2 weeks. In the HT strain, this effect was found in all three post-stress periods studied (24 h, 2 weeks, and 2 months) [9]. Also, changes in the number of glial cells in different hippocampal regions revealed a delayed increase in the number of Iba1+ cells in CA1 (for both HT and LT strains), CA3 (in LT strain), and dental gyrus (in LT strain) in 7 days after PEPS, but not in 1 day or 24 days [16]. These findings suggest that severe cellular selection is detected late after stress and highlight the importance of studying the delayed effects of stress on brain damage because acute effects are not able to predict long-term outcomes.

However, the most significant interstrain differences in post-stress genome instability dynamics were found in PFC. The high-excitable LT strain showed an increase in genome instability 2 h after PEPS, a decrease but still kept it significant from the control level 2 weeks after stress, and a decline to the control level 2 months after the end of the stress procedure (Figure 4D). The increased level of genome instability in the 2-week post-stress period may be the result of the incomplete reparation of DNA damage caused by PEPS, which was already demonstrated 2 h after the stress-ending time point, or the DNA damages can emerge de novo as a delayed consequence of PEPS. On the contrary, an HT strain demonstrated a decrease in the level of DNA damage in PFC cells 2 h after PEPS, but in other post-stress periods studied (2 weeks and 2 months), no significant differences were found between the stressed and control groups (Figure 4A). In previous research, the most contrasting differences between strains in epigenetic modifications (5mC, H3K4me2–3, H3K9me2) were also demonstrated in PFC neurons [9].

In addition, we used the Wistar strain to study the genome responses of the cells of different brain structures to PEPS. We found that the post-stress consequences of DNA instability in all studied regions of the brain (PFC, hippocampus, and amygdala) differ from both the HT and LT strains. Including these data in the discussion about the correlation between hereditary determined nervous system excitability and DNA damage level in brain cells makes it even more complicated. The Wistar strain used for this research had the same threshold of *n. tibialis* sensitivity to electrical stimuli as the LT strain [14]. Because of that, contrast excitability is probably not the only trait that differs between the LT and HT strains. The selection for this characteristic could also fix some other traits in both rat strains; now, we deal with a complex composition of genetic determent differences, which is not yet completely studied. It is the goal of future investigations.

Thus, hereditary-determined excitability defines the dynamics of the response of the genome to stress, but it is important to consider that congenital nervous system excitability is not a constant trait of an individual. Chronic stress itself leads to amygdala hyperexcitability [17] and spontaneous limbic seizures in rats [18]. Also, the hyperexcitability of neurons in the ACC, induced by neuropathic pain, is associated with a depressive state in mice. Furthermore, resilience to depression phenotype, caused by environmental enrichment, is correlated with the decrease in neuronal excitability in ACC [19]. In previous research on our rats’ strains, hereditary-determined high excitability is also correlated with more significant behavioral changes after stress, which was associated with an increase in the number of glial cells in different areas of the hippocampus and a reduction in the neutrophil/lymphocyte ratio [16]. The studies of stress-induced and hereditary hyper- or hypoexcitability complement each other, and their combination can reveal the molecular pathophysiology of a wide spectrum of neurological disorders.

## 4. Materials and Methods

### 4.1. Materials

This study was approved by the Animal Care and Use Committee at the Pavlov Institute of Physiology of RAS (protocol No. 01/16 of 16 January 2023). Male rats of strains with high and low thresholds of sensitivity to electrical stimuli (HT and LT, respectively), originating from Wistar strain, were used for the experiments. The strains are included in the Biocollection of the I. P. Pavlov Institute of Physiology, RAS (No. GZ 0134-2018-0003, patents for selection invention No. 10769 and 10768 issued by the State Commission of the Russian Federation for Testing and Protection of Selection Inventions, registered in the State Register of Protected Selection Inventions on 15 January 2020). The animals were selected in the Laboratory of Higher Nervous Activity Genetics of the I. P. Pavlov Institute of Physiology, RAS [9]. The source material was an outbred population of Wistar albino rats (breeding nursery Rappolovo, Leningrad Region). The selection was carried out according to the value of the threshold of neuromuscular excitability in a test of electric shock irritation (rectangular electrical impulses with a duration of 2 ms) of the tibial nerve, *n. tibialis*. The threshold was estimated as a value of the voltage at which a motor reaction appeared. In the first two generations, full siblings were crossed. Starting from the third generation, intrastrain breeding was carried out in a random order. From the tenth generation on, breeding threshold values reached a plateau. At the same time, the four-fold threshold differences between strains significantly exceeded the intra-strain variability [9].

All animals were kept under standard environmental conditions (23 ± 2 °C; 12 h/12 h dark/light cycle) with ad libitum water and food in the animal care facility at I. P. Pavlov Institute of Physiology, RAS. For each experiment, we took ten (to study short-term effects) or twelve (to study long-term effects) males from each strain: HT, LT, and Wistar, all at the age of five months. The animals (weighing 430.4 g ± 43.7 apiece) of each strain were separated into two groups of five (to study short-term effects) or six (to study long-term effects) animals. Three groups (HT, LT, and Wistar) were stressed, while another three, respectively, served as controls. In total, 4 experiments were carried out, and 122 animals (44 rats of the HT strain, 44 rats of the LT strain, and 34 rats of the Wistar strain (one experiment included only HT and LT rats)) were used.

### 4.2. Exposure to Stressor

The experimental males were exposed daily to prolonged emotional pain stressors (PEPS) for 15 consecutive days (13 min/day). Each animal was placed in a special transparent box and exposed to 12 neutral light stimuli per 10 s according to K. Hecht’s scheme (only 6 of the stimuli were randomly reinforced by a current (2.5 mA, 4 s)) [20]. The interstimulus interval lasted for 1 min. Previous studies have shown that the exposure used contributes to the emergence of persistent behavioral disorders in animals that persist for up to 6 months after exposure [9]. Five or six males (HT, LT, and Wistar) were exposed to the stressor. The same number of animals of each strain was served undisturbed as a matching control. To study short-term consequence of PEPS, 50 rats (20 of HT strain, 20 of LT strain, and 10 of Wistar strain (only to study effects in PFC)) were slaughtered for 1 h (to study effects in amygdala) or 2 h (to study effects in PFC) after the end of the stressing procedure. For research on the long-term effects of stress, 72 rats (24 of each strain) were kept alive for 2 weeks or 2 months after the end of PEPS and slaughtered with a guillotine for instant decapitation to avoid the effects of anesthesia on DNA damage after that time period. The whole amygdala, PFC, and hippocampus were dissected by the same highly experienced experimenter according to the coordinates of the standard rat brain stereotactic atlas [21] and resuspended in standard phosphate-buffered saline (pH = 7.4).

### 4.3. Comet Assay

For the comet assay, cell suspensions were dissolved to a final concentration of ~10^5^ cells/mL. Two protocols of comet assay were used in the research: alkaline (to study short-term effects) and neutral (to study long-term effects). The alkaline comet assay was performed according to the standard procedure with small modifications [22]. Cell suspension (150 μL per specimen) was mixed with an equal volume of a 37 °C 1% solution of low-melting agarose (t_m_ < 42 °C) in microcentrifuge tubes. The obtained 300 μL of the mixture was applied to microscope slides, prepared in advance using the standard method with 1% universal agarose solution base (t_m_ < 65 °C). The mixture was covered with a coverslip (25 × 25 mm), and the microscope slides were placed for 10 min in a refrigerator (t = 4 °C) to harden the gel. All further operations were conducted in the dark or under green light. On the cooled slides, 150 μL of cold lysing solution containing 10 mM Tris-HCl, 2.5 M NaCl, 100 mM EDTA-Na_2_ (pH = 10), 1% Triton X-100 was applied. The slides, covered with parafilm strips, were kept in the refrigerator (4 °C) for 1 h; after that, the lysing solution was washed with phosphate buffer, and the slides were placed into the chamber for electrophoresis (COMPAC-50, Cleaver Scientific, Rugby, UK), and electrophoretic buffer (300 mM NaOH, 1 mM EDTA-Na_2_ (pH > 13)) was added. After 20 min, electrophoresis (EPS-300X Mini-Power Supply apparatus, C.B. S. Scientific, San Diego, CA, USA) was started, which lasted for 30 min (1 V/cm). For the neutral comet assay, the same protocol was performed, with the exception of the final step (instead of NaOH-EDTA-Na_2_ buffer, 90 mM Tris-base, 90 mM boric acid, and 2.5 mM EDTA-Na_2_ buffer was used, and after adding it, electrophoresis was started immediately and lasted 1 h (1 V/cm)). After the procedure had been finished, the preparations were fixed for 5 min in freshly prepared 70% ethyl alcohol–water solution and then were dried at room temperature for 12 h. Two slides from each animal were prepared. The samples were encoded and stained with 1% SYBR Green I (Sigma-Aldrich, St. Louis, MO, USA) for 10 min. The cell nuclei of the amygdala, PFC, and hippocampus (not less than 200 per specimen) were imaged using Axio Scope.A1 (Carl Zeiss, Oberkochen, Germany) and QIClick digital CCD-camera with QCapturePro 7 software (QImaging, Tucson, AZ, USA). The images obtained (not less than 400 suitable for the analysis of the nuclei from each animal) were analyzed using TriTek CometScore^TM^ Freeware v1.5 software (TriTek Corp., Sumerduck, VA, USA).

### 4.4. Statistics

Further statistical analysis was performed using the GraphPad PRISM v. 9.1.0 software package for Windows. After decoding, individual data sets were merged within each experimental group, and summarized data were checked for normality of its distribution (Kolmogorov–Smirnov test). Due to extremely low medians of tDNA content (less than 1% for each experimental group) in the neutral comet assay data (for the study of long-term effects (2 weeks and 2 months after PEPS)), all undamaged cells (containing less than 1% tDNA) were extracted from the analysis. Since the data did not pass the normality test (*p* > 0.1), the Kruskal–Wallis test (with Dunn’s multiple comparison test) was used for the next analysis. For the visualization of DNA destabilization and reparation dynamics, the data are shown as medians with 95% CI for groups that were exposed to PEPS on graphs where the data for each structure are summarized, and the median level of the corresponding control is shown as a Y = 0 line. To summarize the results of all experiments carried out in the research, we also calculated the ratio of medians for all brain structures (amygdala, PFC, and hippocampus) and all rat strains (HT, LT, and Wistar) in all periods after PEPS, which were studied in this research (1 or 2 h, 2 weeks and 2 months) (Figure 4). All data analyzed in each experiment are shown as violin plots, which show medians, quartiles, and the frequency distribution, and are given in the Appendix A.

## 5. Conclusions

After PEPS, the level of DNA damage in cells of the three brain regions studied (PFC, hippocampus, and amygdala) changes significantly, at least in some post-stress time periods for each rat strain used.

(1)In the low-excitable HT strain model of PTSD, we found the following:The level of DNA instability increased immediately after stress in the hippocampus and amygdala and then decreased to the control (in the hippocampus) or even lower (in the amygdala) in 2 weeks, but 1,5 months after, it increased again for both brain areas;In PFC, only 2 h after stress, significant changes (decrease in DNA damage level) were found;(2)In the high-excitable LT strain model of CD, we found the following:
The level of genome instability increased in all three brain regions immediately after PEPS;Delayed changes in the DNA damage level were found only in PFC 2 weeks after the end of the stress procedure;(3)The Wistar strain that was not selected for excitability showed differences from both selected strains’ dynamics of genome instability.

Therefore, there are two main differences in the stress response in two rat strains with contrasting hereditary-determined excitability: the different amplitude and dynamics in genome instability in the hippocampus and amygdala and the changes in the opposite direction of the DNA damage in PFC. The selection to the threshold of *n. tibialis* excitability leads to changes in the level of brain cells’ genome instability, manifesting both in physiological and post-stress conditions, which are implemented into different behavioral patterns, cognitive abilities, and post-stress pathologies in rats.

## Figures and Tables

**Figure 1 ijms-25-00994-f001:**
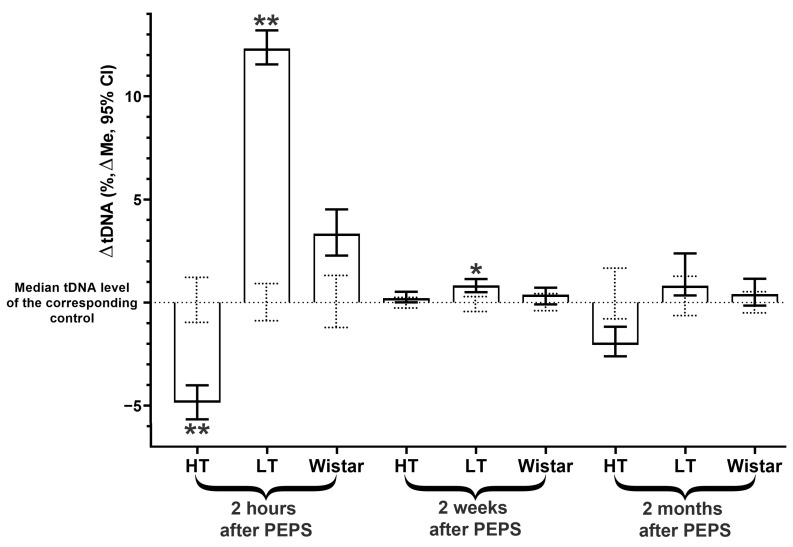
Changes in PFC cell tDNA content of stressed vs. corresponding control groups of different rat strains 2 h, 2 weeks, and 2 months after PEPS. **, *—differences from controls are significant in Kruskal–Wallis test (**—*p* < 0.0001, *—*p* < 0.01). ΔtDNA = tDNAexp − tDNAcontr (full data of all experimental groups are shown in Appendix A). The 95% CIs of the medians of the control groups are shown as dotted lines.

**Figure 2 ijms-25-00994-f002:**
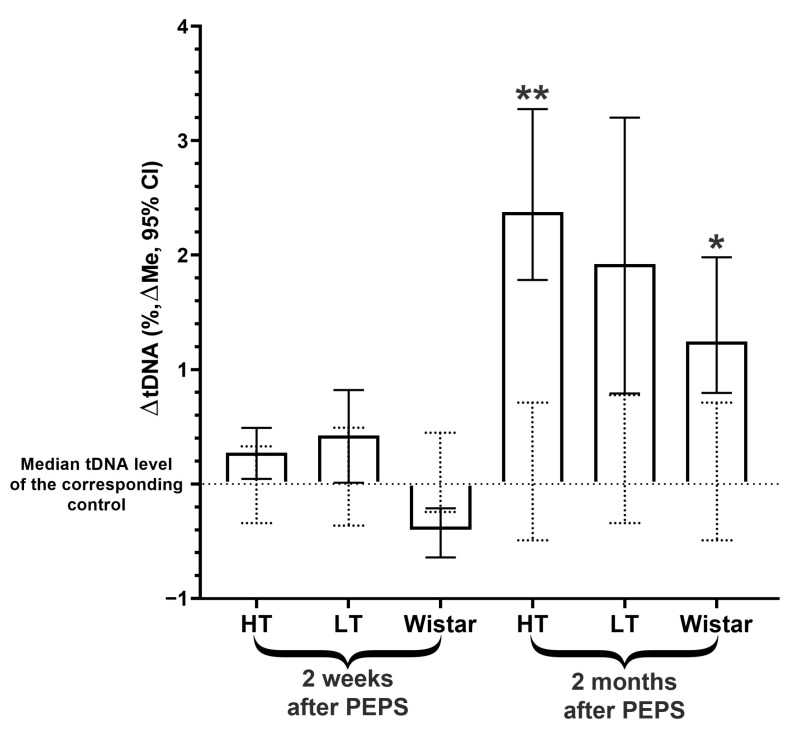
Changes in hippocampal cells tDNA content of stressed vs. corresponding control groups of different rat strains 2 weeks and 2 months after PEPS. **, *—Differences from controls are significant in Kruskal–Wallis test (**—*p* < 0.0001, *—*p* < 0.001). ΔtDNA = tDNAexp − tDNAcontr (Full data of all experimental groups shown in Appendix A). The 95% CIs of the medians of the control groups are shown as dotted lines.

**Figure 3 ijms-25-00994-f003:**
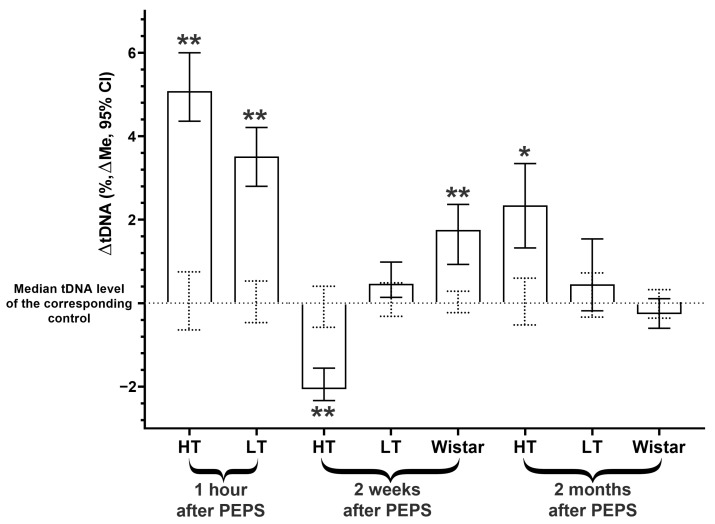
Changes in amygdala cells tDNA content of stressed vs. corresponding control groups of different rat strains 1 h, 2 weeks, and 2 months after PEPS. **, *—differences from controls are significant in the Kruskal–Wallis test (**—*p* < 0.0001, *—*p* < 0.01). ΔtDNA = tDNAexp − tDNAcontr (Full data of all experimental groups shown in Appendix A). The 95% CIs of the medians of the control groups are shown as dotted lines.

**Figure 4 ijms-25-00994-f004:**
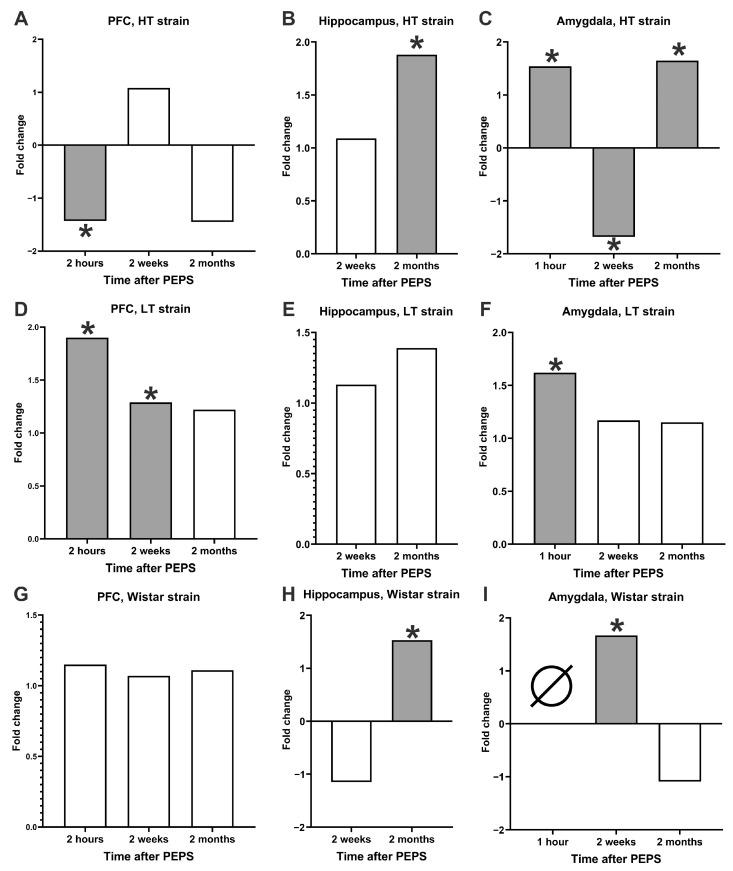
Fold change in the medians of stressed (different times after PEPS) groups from the corresponding unstressed control groups. (**A**,**D**,**G**)—PFC cells; (**B**,**E**,**H**)—hippocampal cells; (**C**,**F**,**I**)—amygdala cells. (**A**–**C**)—HT strain; (**D**–**F**)—LT strain; (**H**–**I**)—Wistar strain. Gray color and *—Difference between the medians of stressed and control groups in Kruskal–Wallis test, calculated for each experiment (all data shown in Appendix A), is significant (*p* < 0.01). ∅—no data available.

## Data Availability

All results obtained in this study are presented in this manuscript.

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
