# Peer review of "Rats Selected for Different Nervous Excitability: Long-Term Emotional–Painful Stress Affects the Dynamics of DNA Damage in Cells of Several Brain Areas"

_ijms, 2024, doi:10.3390/ijms25020994_

Round 1

Reviewer 1 Report

Comments and Suggestions for Authors

In a manuscript submitted for review, the Author described Rats selected for different nervous excitability: long-term emotional-painful stress affects the dynamics of DNA damage in cells of several brain areas

I find the topic of the manuscript interesting and the whole work is thoughtful.

My comments:

- The introduction lacks a formulated hypothesis regarding the experiment presented by the Authors.

- The Authors used 5-month-old animals in the experiment. Was it an intentional effect that? The weight of the tested animals was not given

 - Such animals were selected and not, for example, younger ones?

- Why were only male rats used in the experiment?

- How were animals "slaughtered"? What method was used?

- How were the hippocampus and amygdala extracted? Very generally described methodology on brains extracted

- Were all amygdala nuclei used in the study?

Author Response

Thank you very much for taking the time to review this manuscript. We appreciate your comments, which helped us to improve our article. We considered all your suggestions and added some sentences to provide more details about our current research and important previous results to the manuscript. Please find the detailed responses to your comments below and the corresponding revisions/corrections highlighted in the re-submitted files.

 Point-by-point response to Comments and Suggestions

Comments 1: The introduction lacks a formulated hypothesis regarding the experiment presented by the Authors.

Response 1: We agree with this comment. We added a few sentences and phrases (lines 73-75 and 81-82) in the Introduction section to specify the hypothesis and the goal of this research.

 Comments 2&3: The Authors used 5-month-old animals in the experiment. Was it an intentional effect that? The weight of the tested animals was not given. Such animals were selected and not, for example, younger ones?

Response 2&3: Since the determination of the threshold of excitability and the selection of animals for experimental work are carried out no earlier than the animals reach 3 months of age, accordingly, 5 months is the standard time for experiments to start. We follow this scheme in this work for ease of comparison with previous data. When selecting rats for an experiment, groups are equalized by weight within all strains. All animals used in the research had standard weights for their sex and age (430.4 g ± 43.7 (Mean ± SD)). We added the information about the weight of animals in the Materials section (line 281).

 Comments 4: Why were only male rats used in the experiment?

Response 4: Experiments were carried out only on males due to the presence of estrous cyclicity in females, which would have to be taken into account when organizing an already lengthy and labor-intensive experiment. The goal of this research was not to identify sex differences, although this is an interesting aspect and could be the subject of a special study.

 Comments 5: How were animals "slaughtered"? What method was used?

Response 5: Thank you for pointing out the lack of this information. “The animals were slaughtered with a guillotine for instant decapitation to avoid the effects of anesthesia on DNA damage” (added in lines 300-301).

 Comments 6: How were the hippocampus and amygdala extracted? Very generally described methodology on brains extracted

Response 6: We agree with this comment. We specified the methodology of brain extraction: “Brain regions were dissected by the same highly experienced experimenter according to the coordinates of the standard rat brain stereotactic atlas” (added in lines 302-304).

 Comments 7: Were all amygdala nuclei used in the study?

Response 7: All amygdala nuclei were used in the study (added in line 302).

Reviewer 2 Report

Comments and Suggestions for Authors

To authors

1)     Can the results be extrapolated to any other type of stressor?

2)     "Abstract" section. Please, authors must define abbreviations in the abstract, such as:

- n.tibialis as tibialis nerve.

- PTSD as post-traumatic stress disorder.

- Define the abbreviation HPA (hypothalamus-pituitary-adrenal axis).

- First define the abbreviation PFC (prefrontal cortex) on line 72.

The authors repeat the description of the PTSD abbreviation.

3)     Figure 2. Why is there no data at 1-2 hours after stress?

4)     Also curious are the tDNA values of hippocampal cells in the LT model (2 months after PEPS) and LT control. It is likely that the lack of statistical significance is due to the high variability in its confidence interval.

5)     Lines 128 – 130: “Two months after the end of PEPS in the amygdala, the tDNA content increased only for the HT strain, but for the LT and Wistar strains, no significant differences were shown (Figure 3, Supplementary Figure S8)”.

It is an interesting fact at 2 months: Could it be because the HT strains have greater adaptation to cellular damage and the rest of the strains (LT and Wistar) do not show these changes due to apoptotic processes of the neuronal populations? The loss of significance may point in the direction of resistance due to adaptation, blockages of intracellular signaling pathways, or even alteration in the population number of neurons (proliferation of cell death).

Do the authors have any data on long-term markers of DNA repair, microglial reaction, apoptosis or autophagy?

I would recommend discussing that section further.

6)     Figure 4. Along with paragraph of lines 192 – 203: “Our results reveal that even emotional-painful stress can cause distant damage to the hippocampus and amygdala, penchant for which is not correlated with the level of primary damage. HT strain, where distant brain damage was shown, had less level of DNA instability immediately after stress, than the LT strain, where distant damages of hippocampus and amygdala were not found. This investigation highlights the importance of studying the delayed effects of stress on brain damage because acute effects are not able to predict the longterm outcome. However, the most significant interstrain differences in post-stress genome instability dynamics were found in PFC. LT, a high excitable strain, showed an increase in genome instability 2 hours after PEPS, decreased it but still kept it significant from the control level 2 weeks after stress, and declined to the control level 2 months after the end of the stress procedure (Figure 4D)”.

I thank the authors for creating the graph in Figure 4. This figure provides the reader with the sensitivity of neuronal populations after stress. I ask the authors:

2 months later, some strains show differences and others do not. Could the variable results (major/minor) of tDNA damage, as well as DNA repair, be a determining point for the study of future therapies in some brain regions? According to your results, I observe that the PFC region is essential. But, which brain region should be studied in greater depth because it has a higher risk of neuronal death and results in neurodegenerative disorders (psychiatric and cognitive)?

7)     Supplementary Figures S3 and S7 for PEPSLT group: medians are identified as dotted lines.

8)     Supplementary Figures S5. In the hippocampus and amygdala, the Wistar model seems to be a better model of long-term stress than the LT strain. Which may be due?

Author Response

Thank you very much for taking the time to review this manuscript. We appreciate your comments, which helped us to improve our article. We considered all your suggestions and added some sentences to provide more details about our current research and important previous results to the manuscript. Please find the detailed responses to your comments below and the corresponding revisions/corrections highlighted in the re-submitted files.

Point-by-point response to Comments and Suggestions

Comments 1: Can the results be extrapolated to any other type of stressor?

Response 1: As comet assay reveals an unspecific increase of DNA instability after different stressors (in our studies on rats and mice), it seems possible to extrapolate data obtained here despite some specific peculiarities in stress response shown in different in vivo models. However, additional research is needed.

 Comments 2: "Abstract" section. Please, authors must define abbreviations in the abstract, such as:

- n.tibialis as tibialis nerve.

- PTSD as post-traumatic stress disorder.

- Define the abbreviation HPA (hypothalamus-pituitary-adrenal axis).

- First define the abbreviation PFC (prefrontal cortex) on line 72.

 The authors repeat the description of the PTSD abbreviation. Can the results be extrapolated to any other type of stressor?

Response 2: Thank you for pointing this out. We defined all abbreviations in this article.

 Comments 3: Figure 2. Why is there no data at 1-2 hours after stress?

Response 3:  The data of tDNA content in hippocampal cells at 2 hours after stress have already been published in Shcherbinina et al., 2022 (reference №14). We considered these results in the Discussion section of this article.

 Comments 4: Also curious are the tDNA values of hippocampal cells in the LT model (2 months after PEPS) and LT control. It is likely that the lack of statistical significance is due to the high variability in its confidence interval.

Response 4:  High variability in the study of tDNA content in hippocampal cells two months after PEPS was shown in all three strains of rats used. This fact could lead to a decrease in the significance of the results. However, in all experiments carried out (including some with low variability), significant differences between control and stressed animals typically corresponded to at least 1.5-2 times changes in the median level of tDNA content. For LT strain in hippocampal cells two months after stress only a 1.39-fold change in the median was shown, so it’s unlikely could be significant even if the variability was less. However, only future research with a higher sample size (which we are planning to do) can confirm or refute this result.

 Comments 5: It is an interesting fact at 2 months: Could it be because the HT strains have greater adaptation to cellular damage and the rest of the strains (LT and Wistar) do not show these changes due to apoptotic processes of the neuronal populations? The loss of significance may point in the direction of resistance due to adaptation, blockages of intracellular signaling pathways, or even alteration in the population number of neurons (proliferation of cell death). Do the authors have any data on long-term markers of DNA repair, microglial reaction, apoptosis or autophagy? I would recommend discussing that section further.

Response 5:  We agree that distant DNA damage in 2 months after stress can have many possible explanations. We widened the paragraph describing this data (added lines 156-159 and 161-164) and also added in the Discussion section our previous results, where delayed effects of PEPS on rats’ brains were found (lines 209-216). 

 Comments 6: I thank the authors for creating the graph in Figure 4. This figure provides the reader with the sensitivity of neuronal populations after stress. I ask the authors:

 2 months later, some strains show differences and others do not. Could the variable results (major/minor) of tDNA damage, as well as DNA repair, be a determining point for the study of future therapies in some brain regions? According to your results, I observe that the PFC region is essential. But, which brain region should be studied in greater depth because it has a higher risk of neuronal death and results in neurodegenerative disorders (psychiatric and cognitive)?

Response 6: Most of our previous and current research primarily focuses on the effect of stress on the hippocampus, because of the importance of this brain region to cognitive functions and its role in the formation of psychiatric pathologies. We added some significant results on the effects of PEPS on hippocampal subregions in the Discussion section (lines 209-216). But, as our research shows, it is important to study not only the hippocampus or another region selectively, but also the areas of the brain involved in the response to stress in the aggregate, since understanding the functional relationships between brain regions and regulatory influences is important. We hope that future research will reveal the exact molecular mechanisms, which accompany physiological and pathological changes at the genome level of brain cells to search for potential ways to influence these processes for therapy post-stress pathologies.

 Comments 7: Supplementary Figures S3 and S7 for PEPSLT group: medians are identified as dotted lines.

Response 7:  Thank you for pointing this out. We corrected Supplementary Figures S3 and S7. All medians are shown as solid lines now.

 Comments 8: Supplementary Figures S5. In the hippocampus and amygdala, the Wistar model seems to be a better model of long-term stress than the LT strain. Which may be due?

Response 8:  In the Discussion section, we discussed that HT and Wistar strains have more significant changes in DNA damage level in different brain regions, than the LT strain, but the LT strain previously demonstrated more significant behavioral changes after stress, which was associated with an increase in the number of glial cells in different regions of the hippocampus (lines 250-253). So, it’s hard to decide which of the three strains used in this research is the best model of long-term stress because different results obtained by different methods represent genotype-dependent features of the stress response. The identification of these features is important for the development of personalized approaches to the prediction and correction of post-stress pathologies.

Round 2

Reviewer 2 Report

Comments and Suggestions for Authors

I thank the authors for clarifying my questions, as well as for modifying the manuscript.